# Inter-Comparison Campaign of Solar UVR Instruments under Clear Sky Conditions at Reunion Island (21°S, 55°E)

**DOI:** 10.3390/ijerph17082867

**Published:** 2020-04-21

**Authors:** Jean-Maurice Cadet, Thierry Portafaix, Hassan Bencherif, Kévin Lamy, Colette Brogniez, Frédérique Auriol, Jean-Marc Metzger, Louis-Etienne Boudreault, Caradee Yael Wright

**Affiliations:** 1LACy, Laboratoire de l’Atmosphère et des Cyclones (UMR 8105 CNRS, Université de La Réunion, Météo-France), 97744 Saint-Denis de La Réunion, France; thierry.portafaix@univ-reunion.fr (T.P.); hassan.bencherif@univ-reunion.fr (H.B.); kevin.lamy@univ-reunion.fr (K.L.); 2School of Chemistry and Physics, University of KwaZulu-Natal, Durban 4041, South Africa; 3Laboratoire d’Optique Atmosphérique, Université Lille, CNRS, UMR 8518, F-59000 Lille, France; colette.brogniez@univ-lille.fr (C.B.); frederique.auriol@univ-lille.fr (F.A.); 4Observatoire des Sciences de l’Univers de la Réunion, UMS 3365, 97744 Saint-Denis de la Réunion, France; jean-marc.metzger@univ-reunion.fr; 5Reuniwatt, 97490 Sainte Clotilde de la réunion, France; louisetienne.boudreault@reuniwatt.com; 6Department of Geography, Geo-informatics and Meteorology, University of Pretoria, Pretoria 0002, South Africa; caradee.wright@mrc.ac.za; 7Environment and Health Research Unit, South African Medical Research Council, Pretoria 0001, South Africa

**Keywords:** solar ultraviolet radiation, UV index, UV instruments, clear sky, La Réunion

## Abstract

Measurement of solar ultraviolet radiation (UVR) is important for the assessment of potential beneficial and adverse impacts on the biosphere, plants, animals, and humans. Excess solar UVR exposure in humans is associated with skin carcinogenesis and immunosuppression. Several factors influence solar UVR at the Earth’s surface, such as latitude and cloud cover. Given the potential risks from solar UVR there is a need to measure solar UVR at different locations using effective instrumentation. Various instruments are available to measure solar UVR, but some are expensive and others are not portable, both restrictive variables for exposure assessments. Here, we compared solar UVR sensors commercialized at low or moderate cost to assess their performance and quality of measurements against a high-grade Bentham spectrometer. The inter-comparison campaign took place between March 2018 and February 2019 at Saint-Denis, La Réunion. Instruments evaluated included a Kipp&Zonen UVS-E-T radiometer, a Solar Light UV-Biometer, a SGLux UV-Cosine radiometer, and a Davis radiometer. Cloud fraction was considered using a SkyCamVision all-sky camera and the Tropospheric Ultraviolet Visible radiative transfer model was used to model clear-sky conditions. Overall, there was good reliability between the instruments over time, except for the Davis radiometer, which showed dependence on solar zenith angle. The Solar Light UV-Biometer and the Kipp&Zonen radiometer gave satisfactory results, while the low-cost SGLux radiometer performed better in clear sky conditions. Future studies should investigate temporal drift and stability over time.

## 1. Introduction

Solar ultraviolet radiation (UVR) is known today for its beneficial effects on the biosphere, plants, animals, and humans [1], but also for its negative effects, especially on humans [2]. Most skin diseases are related to UVR exposure [3]. The risk related to UV exposure increases with changes in human behavior, such as increased participation in outdoor activities [4]. UVR is divided into three wavebands: UVA—315–400 nm; UVB—280–315 nm; and UVC—100–280 nm [5]. Surface UVR depends on several atmospheric parameters such as ozone, aerosols, and cloud cover [6], and also geographic parameters, namely altitude and latitude. UV index (UVI) was defined in 1992 by the World Health Organization as a simple tool for public awareness and remains widely used today [7]. UVI starts from zero and increases with UVR intensity. Different thresholds have been defined as a function of risk for human health (i.e., 1–2: low, 3–5: moderate, 6–7: high, 8–10: very high, >11: extreme).

UV solar irradiance is usually measured using a spectro-radiometer and UVI is calculated via a standard formula [8]. Broadband UVR radiometers are also used with a spectral response that is adapted to the UV erythemal (sunburn) action spectrum. Many instrument uncertainties have to be considered [9]. Moreover, UVR instruments have to be regularly checked. 

Reunion Island (55°E, 21°S) is a tropical island situated in the Western Indian Ocean (Figure 1) and is exposed to extreme UVR all year around. UVI is generally in the range of 0 to 16 and can exceed 20 under certain conditions, such as at high altitude or cloud diffusion [10]. Therefore, UVI measurement, public awareness, and prevention campaigns are very important for Reunion Island and in tropical regions in general. The Laboratoire de l’Atmosphère et des Cyclones (LACy) is located in Saint-Denis (Figure 1) at the University of Reunion, where a Bentham DTMc300 spectroradiometer has been in use since 2009. The uncertainty on UVI is about 5%, with a coverage factor of k = 2 [11]. The instrument is affiliated with the Network for the Detection of Atmospheric Composition Change (NDACC). There are other Bentham DTMc300 instruments in operation around the world. Two Bentham DTMc300 instruments have been in operation in metropolitan France at Villeneuve d’Ascq (in the North) and Observatoire de Haute-Provence (in the South) since 2009 with similar uncertainty [11]. In Italy, Aosta Valley, a Bentham DTMc300 has recorded spectral UV irradiance since 2006 with 5% bias to QASUME (Quality Assurance of Spectral Ultraviolet Measurements in Europe) through the Development of a Transportable Unit [12]. Similar results were found with DTMc300 instruments operating in Germany, Great Britain and New Zealand [13].

There are currently various instruments dedicated to the measurement of solar radiation in the ultraviolet (UV) band of 280 nm to 400 nm. These instruments are available at different price ranges; however, price does not guarantee quality of the measurement. The present study aimed to compare, at the same site with high level of UVR and under the same experimental conditions, a set of UVR sensors commercialized at low to moderate costs. The objective was to assess their individual performance and the quality of the respective measurements. To this end, we set up an inter-comparison campaign bringing together five instruments (including the Bentham DTMc300 spectrometer) for evaluation over a continuous 12-month period from March 2018 to February 2019. The datasets collected over the study period were evaluated, taking into account the cloud fraction data obtained by a co-localized all-sky camera.

## 2. Materials and Methods

### 2.1. Materials

#### 2.1.1. Spectroradiometer Bentham DTMc300

The reference instrument for this study was a high grade double monochromator Bentham DTMc300 (Figure 2) provided by Bentham Instrument Ltd. Co. (Reading, England, United Kingdom), hereafter referred to as BT, operated by the OPAR (Observatoire de Physique de l’Atmosphère de la Réunion) since January 2009. The BT is affiliated with the NDACC. The instrument is enclosed in a thermally-stabilized box and records global irradiance spectra in the 280–450 nm wavelength range every 15 min in its user configuration, with a wavelength scan duration of about five minutes. The BT is calibrated every 3 months with a 150W lamp and a 1000W quartz tungsten halogen lamp from the National Institute of Standards and Technology (NIST). Wavelength misalignment correction is done via a software developed at the Laboratoire d’Optique Atmosphérique [14] using a cosine correction function.

The erythemally-weighted UVR is obtained by integrating the global irradiance in the 280–400 nm wavelength range weighted by the erythemal action spectrum (Commission Internationale de l’Eclairage (CIE) S007-1998 [15]). The UVI is calculated following standard formulae. The instrument uncertainty of UVI is about ±5%, with a coverage factor of k = 2 [11]. 

In 2013, during a QASUME campaign [16], a BT/QASUME ratio of −5% to 0% was found [17]. A recent comparison between BT and UVI obtained by modelling showed ±5% difference [18]. However, since 2009, there have been some gaps in the data due to technical problems and prolonged maintenance delays.

The UVI data measured by BT are presented in Figure 3**.** There was satisfactory time-coverage data, as only 12% of data were missing for the period March 2018 to February 2019. The UVI range during the inter-comparison exercise was 0 to 16. The seasonal maximum appeared during mid-summer (January) with a UVI up to 16, while the UVI was around 8 during winter.

The BT cost is ~50.000 € and an estimation of the calibration cost is ~1000 €/year for seven full days of work (four calibrations per year). These costs do not take into account unexpected maintenance operation.

#### 2.1.2. Radiometer Kipp&Zonen UVS-E-T

A Kipp&Zonen UVS-E-T (Figure 2) was used for the inter-comparison campaign, referred to hereafter as KZ. It is a moderate-cost (~3000 €) broadband radiometer recording erythemal irradiance in the 280–400 nm wavelength range. The erythemal action spectra used for UVI calculation is defined by the CIE (CIE S007/E-1998). Erythemal irradiance is recorded every minute. However, final data are averaged using five erythemal irradiance records and are given every five minutes. According to Gröbner et al. [19] the KZ uncertainty is ±7%. 

The radiometer was calibrated during the international UV filter Radiometer Comparison in summer 2017 by Physikalisch-Meteorologisches Observatorium Davos/World Radiation Center (PMOD/WRC) in Davos [20]. A calibration factor was given, with an ozone and zenith angle correction. Erythemal irradiance was calculated using Equation (1) below [21]:(1)$$E_{CIE}=\left(U-U_{dark}\right)\cdot C\cdot f_{n}\left(\theta ,TO_{3}\right)\cdot Coscor\left(\theta \right)\;,$$
where *E_CIE_* is the erythemal weighted irradiance, *U* is the raw signal of the instrument, *U_dark_* is the dark offset, C is the calibration factor, determined for the solar zenith angle *θ* = 40° and the total column of ozone *TO*_3_ = 300 DU, *f_n_* is a function of *θ* and *TO*_3_, *COSCOR* is the cosine correction function. It is a dimensionless factor that rectifies the mismatch between the actual angular response of a radiometer and the ideal behaviour expected, given by the cosine law.

This radiometer is part of the UV-Indien UVR Observation Network in the Western Indian Ocean [22].

#### 2.1.3. Radiometer Solar Light UV-Biometer Model 501

The SL501 UV-Biometer (Figure 2) is a moderate-cost (~5000 €) broadband radiometer manufactured by Solar Light Pty Ltd, referred to hereafter as SL. UVR is recorded between 280 to 340 nm with a 1-minute sampling rate. Since the SL wavelength range differs from the UVI standard, a spectral correction depending on total ozone and solar zenith angle is applied from a generic table for this specific instrument [23]. Data are recorded in [MED/h] units, where 1 MED (minimal erythemal dose) is 210 J·m^−2^. UVI was calculated using Equation (2):
(2)$$UV\;index=UVd\left[MED\cdot h^{-1}\right]\cdot \frac{210\left[J\cdot m^{-2}\right]\cdot 40\left[m^{2}\cdot W^{-1}\right]}{3600\left[s\right]}\;,$$

According to Gröbner et al. [19] the SL uncertainty is ±7%. The instrument is calibrated in intensity and corrected to solar zenith angle and total ozone. This calibration was done during the International UV Filter Radiometer Comparison in summer by PMOD/WRC at the World Calibration Centre in Davos [20] (See Section 2.1.2).

#### 2.1.4. Radiometer SGLux UV-Cosine

The UV-Cosine sensor (hereafter referred to as SG) from SGLux Company is a low-cost (~250 €) broadband radiometer integrating erythemal UVR following the ISO 17166 erythema action spectra in the 280–400 nm wavelength range (Figure 2). The solar zenith angle correction and the calibration factor are provided by the manufacturer and applied by Reuniwatt. Data are recorded every minute.

#### 2.1.5. Radiometer Davis

The DAVIS UV sensor referred to as DV (Figure 2) operates on the Vantage Pro 2 meteorological station from DAVIS Company. It is a low-cost (~200 €) radiometer recording UVR in the 280–360 nm wavelength range. UVI, dose, and cumulative dose are provided every second. The sensor was calibrated by the manufacturer.

#### 2.1.6. Sky Camera

Reuniwatt’s *SkyCamVision* (hereafter referred to as SCV) is an all-sky camera system acquiring hemispherical images of the sky vault in the visible range (380–440 nm) at 1-minute intervals. The system is equipped with a CMOS sensor (1600 × 1200 pixels resolution) mounted with a fisheye lens. More information about the system specifications is available at the website: http://www.reuniwatt.com/.

The cloud fraction calculation using this device involves a multi-step procedure [24]. A cloud segmentation algorithm is applied to classify the pixels as either: (1) clear sky; (2) thick cloud; (3) thin cloud; or (4) sun. The classifier is based on a Random Forest algorithm [25], which is programmed beforehand using as inputs a number of pixel features from manually annotated images using both the RGB and HSV colour spaces. Once the classification is achieved, the sun and clear sky indices (1–2) and the thick and thin cloud indices (2–3) are merged together into a binary cloud cover: single clear sky (0) and cloud (1) classes. The overall cloud fraction is then calculated from a geometrically calibrated image. The raw image is undistorted onto a flat plane of reference perpendicular to the zenith, and each pixel of the cloud cover is weighted by its solid angle view [26].

This camera is part of the UV-Indien UVR Observation Network in the Western Indian Ocean [22].

#### 2.1.7. TUV Model

We used the Tropospheric Ultraviolet Visible (TUV) radiative transfer model version 5.3 [27] to obtain modelled clear-sky outputs. The radiative transfer scheme in TUV is used to solve the radiative transfer equation in pseudo-spherical 8-stream discrete ordinates [28]. The following parameters were modified in the model in order to reproduce the UVI measurements with site-specific climatology:–extraterrestrial spectrum (ETS),–solar zenith angle (SZA),–total ozone column amount (TO3),–total nitrogen dioxide (TNO2),–ozone profile (OP),–temperature profile (TP),–aerosol optical thickness (AOT) at 340 nm–aerosol Ångström exponent (α) between 340 and 440 nm,–single-scattering albedo (SSA) [29,30],–ground surface albedo (ALB) and–altitude (Z).

The ETS used was from Dobber et al. [31]. Similar to McPeters et al. [32], a monthly climatology of ozone and temperature profiles was derived from local ozone soundings and Microwave Limb Sounder (MLS) satellite measurements. TO3 and TNO2 used are from a Système d’Analyse par Observation Zénitale (SAOZ) instrument. AOT and α data used were from the Aerosol Robotic Network (AERONET). As demonstrated by Dubovik et al. [33], single-scattering albedo (SSA) from the CIMEL sun photometer is not usable when the AOT is lower than 0.3, which was almost always the case here. As proposed by Takemura et al. [29] and Lacagnina et al. [30], a fixed SSA of 0.95 was set. As described by Corrëa et al. [34], UVI doses can be reduced by 10 to 30% for a lower SSA of about 0.70 which indicates the presence of strongly absorbing aerosols. These aerosols are observed for small areas, limited time periods, and during specific events, such as biomass burning emissions or fires with incomplete biomass burning episodes. As established by Koelemeijer et al. [35], surface albedo was taken to be constant at 0.08. According to Koepke et al. [36], the UVI modelling error is approximately 5% for a coverage factor of 2.

Figure 4 shows the monthly mean UVI captured by the TUV model compared to those of BT by using all data (Figure 4a; left panel) and clear sky data (Figure 4b; right panel). As expected, comparisons are better with clear-sky filtered observation data (Figure 4b) as the model outputs are made without taking into account cloud cover. However, the differences on the left panel (Figure 4a) are greater during the summer months. This result and the method of filtering are discussed later. 

### 2.2. Methods

The objective of the instrument comparison exercise was to evaluate their performance and quality of measurement. All of the instruments, i.e., BT, KZ, SL, SG, DV, and SCV, were co-localised on the Moufia Campus of Reunion University in Saint-Denis (20.9°S, 55.5°E, 85 m ASL), with the Bentham spectro-radiometer (BT) as the reference. Solar zenith angle range is from 0° to 90° during summer and from 45° to 90° during winter. The comparisons were performed during a one-year period, from 1 March 2018 to 28 February 2019. As the instruments evaluated during the inter-comparison campaign do not measure at the same time resolution, the corresponding recorded time-series were interpolated, according to the zenith angle, with a one-degree step.

Cloud fraction (CF) data calculated from the all-sky camera SCV were used. Yearly statistics are presented to describe the cloud cover conditions over Saint-Denis. The instrument inter-comparison was performed only for clear sky conditions. Cloud fraction data were analysed in two ways to determine a clear sky threshold: 

(1) Lamy et al. [18] showed that TUV clear sky outputs could be compared to Bentham clear sky outputs on the same site with differences of less than 5% when TUV was correctly set up. From this result we compared the TUV outputs with the appropriate setting and the Bentham over the reference period in order to determine the cloud fractions (as seen by the camera) corresponding to a difference in UVI of less than 5%. Associated cloud fraction was analysed to determine the cloud fraction clear-sky threshold; 

(2) Clear sky UVI was determined by using the filtering method developed by Bodeker and McKenzie [37] based on the geometrical form of the daily UVI curve. Three tests were done on BT UVI to determine clear-sky days: (a) a correlation test; (b) a monotonicity test between morning and afternoon UVI; and (c) a comparison of the daily maximum to the climatology. Associated cloud fraction was analysed to determine the cloud fraction clear-sky threshold.

The statistical analysis was initially carried out on the data measured over the entire study period, after which analyses were done per month. A statistical analysis was also done as a function of solar zenith angle. Statistical tools for the analysis were determination coefficient (r^2^), relative difference (RD), relative standard deviation (RSD), root mean square error (RMSE), as shown in Equations (4)–(7), and box diagrams.
(3)$$DIFF_{\left[X\right],i}=\frac{UVI_{\left[X\right],i}-UVI_{\left[BT\right],i}}{UVI_{\left[BT\right],i}},$$
(4)$$r^{2}=\frac{{\left(\sum_{i=1}^{n}\left(UVI_{\left[BT\right],i}-\bar{UVI_{\left[BT\right]}}\right)\left(UVI_{\left[X\right],i}-\bar{UVI_{\left[X\right]}}\right)\right)}^{2}}{\left(\sum_{i=1}^{n}{\left(UVI_{\left[BT\right],i}-\bar{UVI_{\left[BT\right]}}\right)}^{2}\right)\left(\sum_{i=1}^{n}{\left(UVI_{\left[X\right],i}-\bar{UVI_{\left[X\right]}}\right)}^{2}\right)},$$
(5)$$RD=100*\frac{1}{n}\sum_{i=1}^{n}DIFF_{\left[X\right],i},$$
(6)$$RSD=100*\frac{\sqrt{\frac{1}{n-1}\sum_{i=1}^{n}{\left(DIFF_{\left[X\right],i}-\bar{DIFF}\right)}^{2}}}{\bar{DIFF}},$$
(7)$$RMSE=\sqrt{\frac{1}{n-1}\sum_{i=1}^{n}{\left(UVI_{\left[X\right],i}-UVI_{\left[BT\right],i}\right)}^{2}},$$
where *n* is the number of observations and [*X*] the instruments compared (KZ, SL, SG and DV).

## 3. Results and Discussion

### 3.1. Cloud Cover Conditions over Saint-Denis

Cloud cover and its daily cycle contribute to the variability of solar surface UV radiation at different timescales. As previously mentioned, in order to understand this variability an all-sky camera was installed in the vicinity of UV instruments over the roof of the Physics Department, Moufia Campus. The all-sky camera has been operating continuously and allows retrieval of cloud fraction values on a daily basis. Figure 5a shows the daytime statistics obtained (hourly median values) of cloud fraction as derived from one year of sky imaging over the study site. Overall, the values of cloud fractions remained relatively low in the morning (below 30%, until 11 am), while they exceeded 50% from 1 pm, with the maximum (68%) around 4 pm. This was consistent with the prevailing meteorological conditions and the geographical location of the study site. Indeed, as shown in Figure 5b, Reunion is an island located in the southwestern Indian Ocean. It is characterized by mountainous terrain (the highest peak, Piton des Neiges, is 3069 m) and is subjected to the southeast trade winds, which implies cloud developments on the windward side, with an overflow of clouds in the early afternoon on the opposite side, where the study site is located. Such cloud development is typically characteristic of isolated islands in the tropics [38]. There is also a difference in cloud cover between summer and winter months. Reunion Island is located in a tropical region, which implies that there is a strong cloud cover during the summer months [39,40]. The strong cloud cover which occurs during the wet season (from November to March) introduces more bias, as can be seen in Figure 4b. However, the bias in clear-sky conditions remains a factor within the BT uncertainties. These results are consistent with findings from Lamy et al. [18] for the time period 2009–2018.

### 3.2. Clear Sky Filtering

We applied a statistically determined threshold to select clear sky UVI measurements:

In the first method proposed, we compared UVI from BT and TUV model outputs, and detected cloud fractions (CFs) when the difference was less than 5%, as reported in Lamy et al. [18]. Figure 6a shows the distribution of the clear-sky CFs selected. The data range of selected CFs was from 2% to 100% and represented 24% of all data. There were no observations between 0 to 2% and an obvious concentration of observations between 2% to 6%, even though the median was 9%. The weighted mean was 19%. 

For the second method proposed [37], the analysis of the clear-sky cloud fraction observations (Figure 6b) showed the same pattern as Figure 6a with a median at 15% and a wider dispersion, while the weighted mean was 29%. The disadvantage of using this method of filtering is that only full days of clear-sky are detected and therefore partial clear-sky periods during the day could not be taken into account. Therefore, this method was not used.

The dispersion of the observations in both cases from the median to the maximum represents the limit of the cloud fraction data. Indeed, cloud fraction data does not indicate whether the sun is hidden by clouds. The cloud cover can be important, but the UVR can be maximal if the sun is not hidden by clouds. The UVR can also be increased in the case of scattered cloud cover [10,41].

It was decided that the results from the first method, based on the difference between BT UVI and TUV clear sky UVI to determine clear sky measurement, would be used. A threshold of 20% of the cloud fraction measured by the all-sky camera was applied, which corresponded (approximately) with the statistical average shown above, and up to about 70% of the clear sky conditions detected. Figure 7 shows the UVI as recorded by BT and CF, as recorded by SCV on 19/09/2018. 

### 3.3. Inter-Comparison

The present work aimed to assess the quality of a set of UV sensors selected as low- to moderate-cost instruments. This was achieved through a continuous one year inter-comparison campaign with a BT spectrometer selected as the reference instrument. The results of this comparison are statistically summarised below in Table 1 and are shown in Figure 8 and Figure 9. From the r² values (in the 3rd column of Table 1), the four UV instruments show a very good correlation with the BT (99%). It is expected that these correlations are very high because the same annual UVI course dominates the data sets and all instruments were co-located. 

It can be seen from plots of Figure 8a,b that data from the KZ radiometer is very closely aligned with that of the BT spectrometer. Indeed, the RD between the two instruments is as low as 1.1%, with a median RD value of 2.6%. The scatter plot (Figure 8a) and the RD distribution (Figure 8b) show a low dispersion of the deviations. Furthermore, Figure 9b shows a low dependence of the KZ RD with respect to the zenith solar angle, i.e., averages for several solar zenith angle ranges show a decreasing RD with increasing zenith angle within ±6%. As expected, and shown in plot of Figure 9a, this low zenith angle dependence results in low variations of the KZ RD over time (RD averages for several months), within ±4%. Nevertheless, this small but not negligible zenith angle dependence shows that the cosine correction introduced in the calibration could be improved.

With regard to UVI measurements from SL and SG radiometers, a very close alignment with BT is noted. They show similar narrow RD dispersions, with low RD values of 3.1% and 1.4%, and median values of the relative differences at 1.7% and 1.2%, respectively (see plots of Figure 8d,f). When one considers the evolvement of RD over months (Figure 9c,e) as a function of the solar zenith angle, by looking at the RD for several solar zenith angle ranges (Figure 9d,f), the mean RD remained almost constant for both instruments. However, the small positive RD obtained for the SL radiometer between June and October (around +5%), when there is virtually no dependence on the zenith angle, should be noted.

The Davis radiometer showed an RD of 14.2%, with a 13.3% median compared to BT. The RD dispersion is higher than the other instruments (Figure 8g,h). The DV was not corrected for ozone or solar zenith angle which may explain the RD. 

Figure 9h clearly shows a dependence of the RD on the zenith solar angle, ranging from 0% at low solar zenith angles and increasing up to 30% at 70° solar zenith angle. It should be noted that the RD is very small at low solar zenith angles, when UV radiation is most intense. There is also a seasonal oscillation of the RD during the comparison period due to the dependence of the RD on SZA, in the order of ±4% around the mean RD, with a minimum in summer (DJF) and a maximum in winter (JJA). This is consistent with the annual evolution of mean SZA.

Overall, the panels on the left of Figure 9 highlight the reliability of the instruments over time, despite a +14% shift for the DV. The right side of Figure 9 shows a slight dependence on solar zenith angle except for the DV, which was not corrected for solar zenith angle.

No dependence on solar zenith angle was found for KZ, SL and SG. The same behaviour can be expected if these sensors are used in other latitude sites. However, RD depends on solar zenith angle for the DV. Therefore, the global RD depends on the solar zenith angle range of the site of use. At low latitude, the same RD behaviour as for the Reunion site is expected. However, with regard to latitudinal variation, the RD could increase with latitude, polewards from the equator. 

Several Kipp&Zonen UVS-E-T and Solar Light SL501 have also been compared in other comparison campaigns in El Arelosillo, Spain [42] and Davos, Switzerland [19,20] using QASUME as reference. The results showed good compliance with the reference, with a RD less than ±10%, except in a few instances. The results shown here are fairly consistent with these previous studies, even though the UV levels reported here were much higher than those of the previous studies.

The SGLUX UV-Cosine and the Davis Radiometer are two new UV sensors. There is no comparison of high grade UV instruments in the literature as yet.

## 4. Conclusions

In conclusion, an inter-comparison of four radiometers to a spectroradiometer Bentham DTMc300 (as reference) was performed. Bentham DTMc300 is a high-grade instrument, affiliated WITH NDACC. Comparison with TUV model showED less than 5% difference. All of these instruments were installed on the same instrumental platform at the University of Reunion and operated simultaneously between February 2018 and March 2019. The purpose of such a study was to qualify these UV sensors under high irradiances, such as those observed in the tropics and on Reunion Island in particular. Of the instruments compared, some were moderately costly (3000–5000 €) while others were fairly inexpensive (a few hundred euros).

The clear sky filtering was carried out using an original method implemented at LACy, based on the measurement of the cloud fraction from an all-sky imaging camera. Clear sky time was characterised by a maximum threshold of 20% cloud fraction. Typical tropical cloud conditions are illustrated by cloud fraction data: clear sky mainly in the morning, with an increase of cloud fraction in the afternoon. Similar daily cloud cover conditions may be found at different tropical sites.

The results show that two medium-cost instruments (KZ and SL) give very satisfactory results with regards to the RD in comparison with the BT. Nevertheless, a small zenith angle dependence remains with KZ and the cosine correction introduced in the calibration could be improved.

A clear dependence on solar zenith angle was found for the DV, with a 14% RD. A specific calibration should be done before using this instrument with a solar zenith angle correction. 

It was also found that the low-cost SGLux UV cosine instrument performed quality measurements with an average clear sky RD of less than 1.5% compared to our reference measurement and a low dispersion (max. 10%).

In the future, it will be interesting to investigate the temporal drift of these instruments after different lengths of operating time, as well as in different geographical locations, and to check whether even low-cost instruments like SGLux UV Cosine remain reliable over time.

In addition, it would be interesting to develop integrated low-cost measurement solutions based on the SGLux UV cosine in order to increase the number of observation sites on Reunion Island and in neighbouring Indian Ocean countries.

## Figures and Tables

**Figure 1 ijerph-17-02867-f001:**
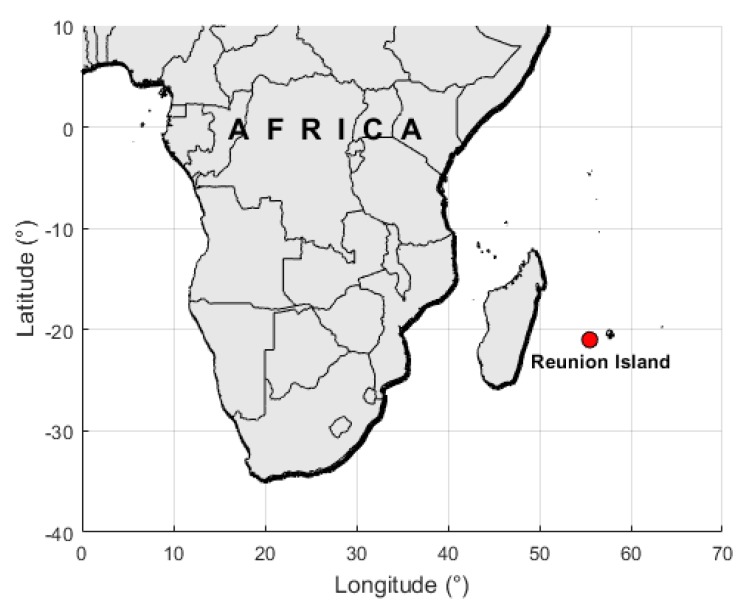
Geographical location of the study site, Saint-Denis, La Réunion.

**Figure 2 ijerph-17-02867-f002:**
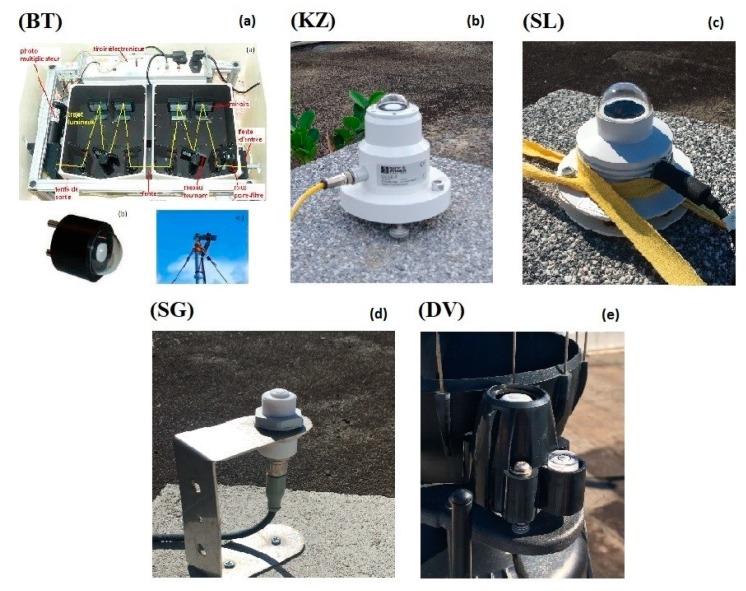
Images of the UV instruments evaluated during the inter-comparison campaign held at Reunion University. (**a**): Bentham DTMc300; (**b**): Kipp&Zonen UVS-E-T; (**c**): Solar Light UV-Biometer Model 501; (**d**): SGLux UV-Cosine; (**e**): Radiometer Davis.

**Figure 3 ijerph-17-02867-f003:**
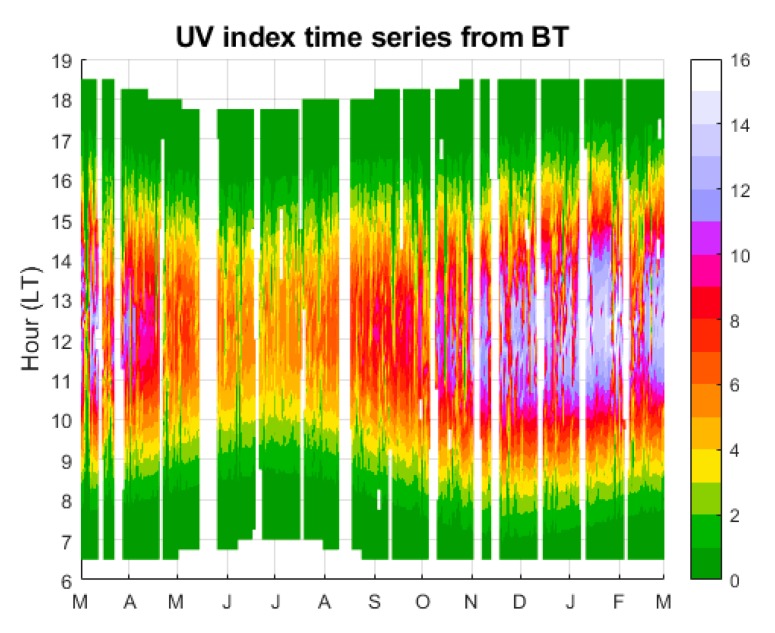
Time series UV index measured by Bentham spectro-radiometer (BT) during the period of comparison (March 2018 to February 2019).

**Figure 4 ijerph-17-02867-f004:**
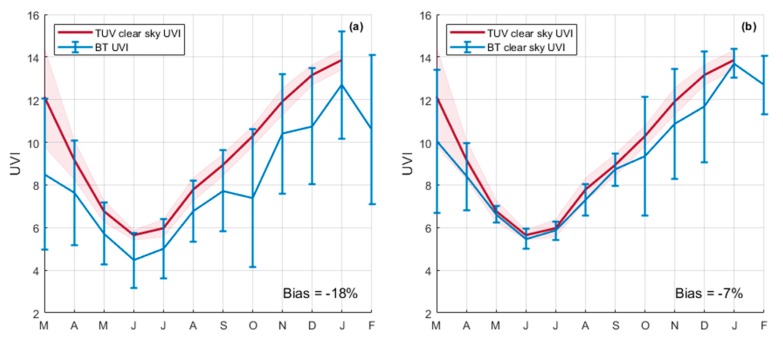
Comparison of UVI between the TUV model (red) and BT (blue). Standard deviation is represented by red shading for TUV data and by vertical bars for BT data. Plot (**a**) represents the comparison by using all data, while plot (**b**) represents clear sky data only.

**Figure 5 ijerph-17-02867-f005:**
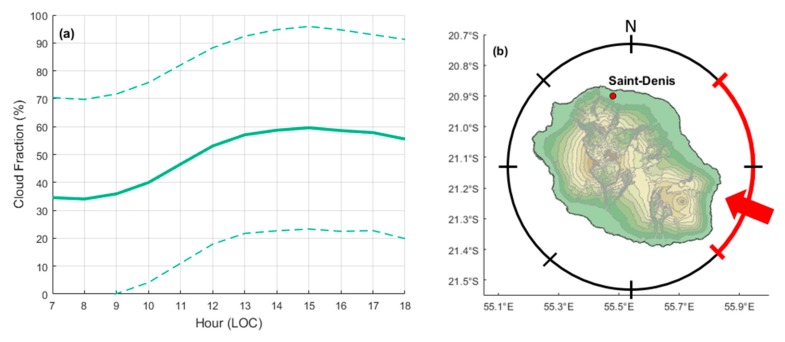
(**a**) Cloud fraction statistics over Saint-Denis. The green solid line shows the mean cloud fraction per hour, and the dashed lines represent one standard deviation. (**b**) Reunion island map and the dominant trade winds. The red arrow shows the main direction of the trade winds with a possible variation illustrated by the red arc. The study site (Saint-Denis) was located in the North and is indicated by a red dot.

**Figure 6 ijerph-17-02867-f006:**
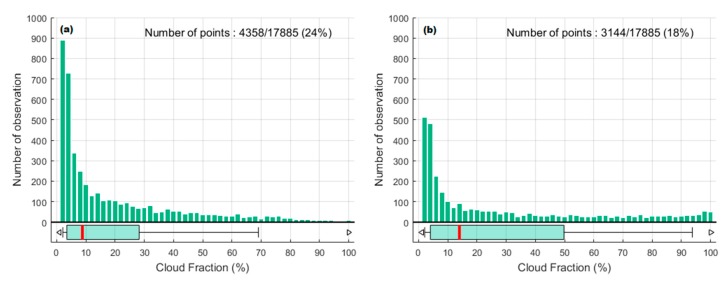
Distribution of clear sky cloud fraction (CF) data from the all sky camera. The green diagram boxes at the bottom represent the quantiles 0, 0.05, 0.25, 0.50, 0.75, 0.95, 1. (**a**) shows the clear-sky conditions in CF by using data selected when the difference between UVI from BT and clear sky UVI from TUV model are less than 5%. (**b**) shows the clear-sky conditions in CF by using the Bodeker and McKenzie [37] filtering method.

**Figure 7 ijerph-17-02867-f007:**
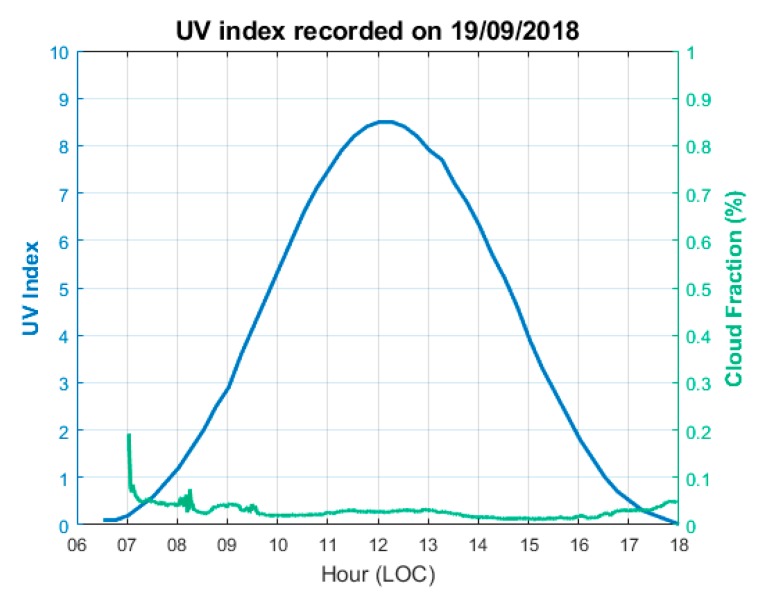
UVI (BT) and CF (SCV) distribution on 19/09/2018 at Saint-Denis. The blue line represents the UVI and the green line represents the CF.

**Figure 8 ijerph-17-02867-f008:**
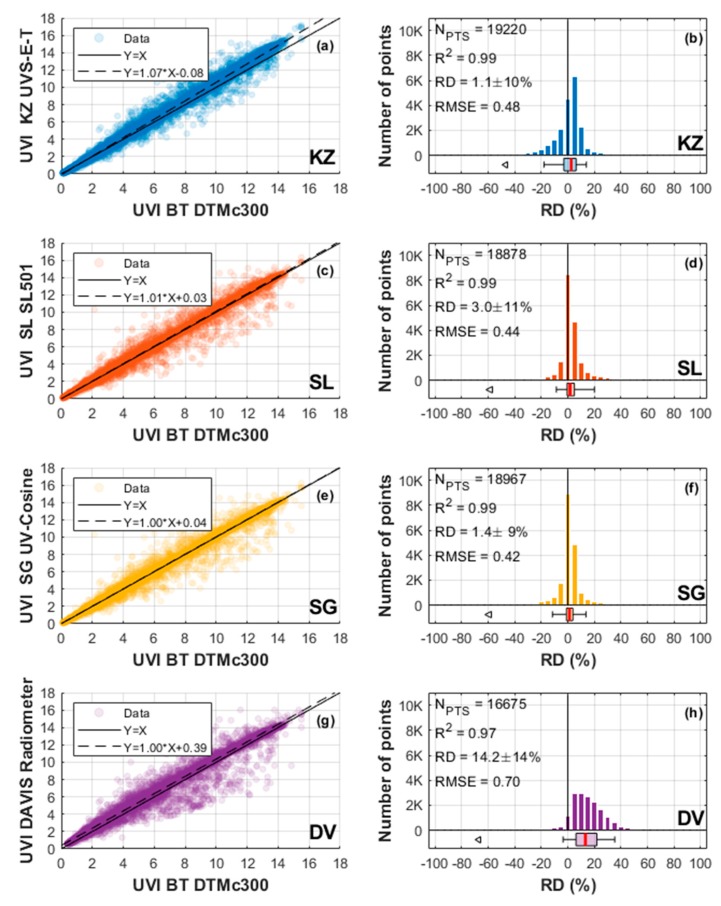
Comparison between BT and all of the instruments under evaluation. The left column (**a**,**c**,**e**,**g**) shows a scatterplot specific for each instrument with a linear fit; the right column (**b**,**d**,**f**,**h**) shows the RD distribution with statistics. The box diagram shows the 5e, 25e, 50e, 75e, 95e percentile and the arrows show the minimum and the maximum when they appear in the X-axis range.

**Figure 9 ijerph-17-02867-f009:**
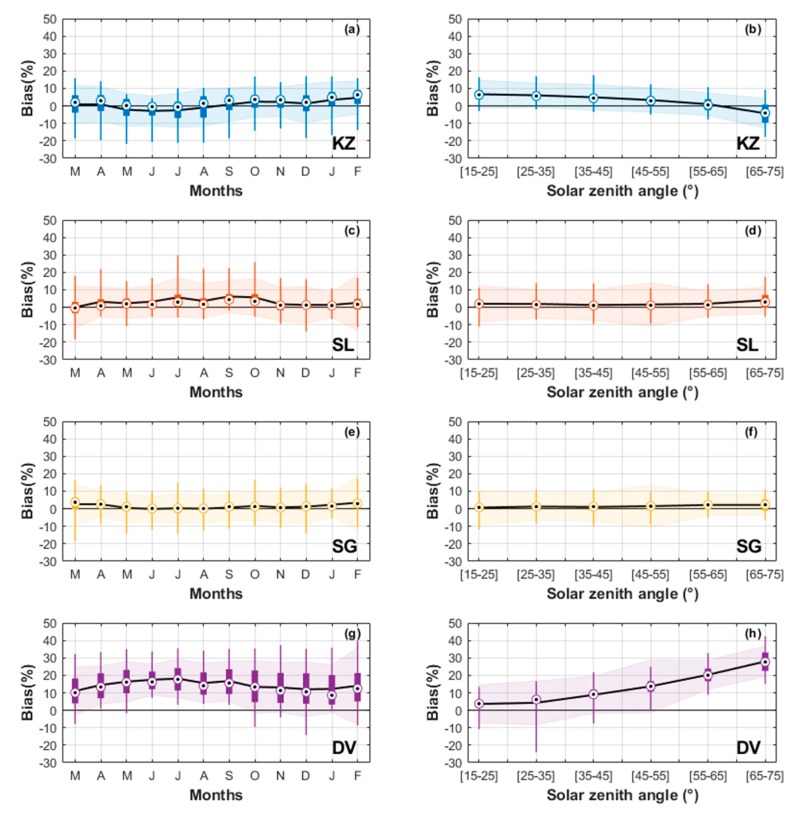
The left column (**a**,**c**,**e**,**g**) shows the evolvement of the RD as a function of time, and the right column (**b**,**d**,**f**,**h**) shows the evolvement of the RD as function of solar zenith angle. The black solid line represents the mean RD and the shaded surface one RSD. The box diagram shows the 5e, 25e, 50e, 75e, 95e percentile.

**Table 1 ijerph-17-02867-t001:** Summary of comparison statistics between BT and all instruments. RD, relative difference; RSD, relative standard deviation; RMSE, root mean square error; KZ, Kipp&Zonen UVS-E-T; SL, SL501 UV-Biometer; SG, SGLux UV-Cosine sensor; DV, DAVIS UV sensor.

	Number of Points	r² (%)	RD (%)	RSD (%)	RMSE	Quantile (%)				
						0.05	0.25	Median	0.75	0.95
KZ	19220	99	1.1	10.2	0.5	−17.9	−2.9	2.6	6.3	14
SL	18878	99	3.1	10.6	0.4	−8.6	−0.5	1.7	4.9	20.1
SG	18967	99	1.4	9.4	0.4	−11.6	−1	1.2	3.8	13.7
DV	16675	99	14.2	14.5	0.7	−3.5	6.2	13.3	21.9	35.4

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
