# Peer review of "Inter-Comparison Campaign of Solar UVR Instruments under Clear Sky Conditions at Reunion Island (21°S, 55°E)"

_ijerph, 2020, doi:10.3390/ijerph17082867_

Round 1

Reviewer 1 Report

Intercomparisons of UV measuring instruments are important in order to ensure the quality of the data on UV irradiances. Based on this the manuscript is publishable. Several minor comments are:

1) Provide the range of noon solar zenith angles for March 2018 to February 2019 at the location of the intercomparison.

2) Equation 1 has "Coscor". What is this and it needs to be defined.

Author Response

Dear Reviewer,

First of all, I would like to thank you for your work on our paper. Our response is structured based on your review. We identified two comments, referred hereafter as C1 and C2. Our responses are labelled R1 and R2. To make the reading of the revised manuscript easier, all edits are highlighted in yellow.

With kind regards,

Jean-Maurice CADET

Intercomparisons of UV measuring instruments are important in order to ensure the quality of the data on UV irradiances. Based on this the manuscript is publishable. Several minor comments are:

C1. Provide the range of noon solar zenith angles for March 2018 to February 2019 at the location of the intercomparison.

R1. The range in solar zenith angle has been provided. (see Lines 220-221)

C2. Equation 1 has "Coscor". What is this and it needs to be defined.

R2. For more clarity, the definition of the COSCOR function has been modified in the revised version of the manuscript, as follows: COSCOR is the cosine correction function. It is a dimensionless factor that rectifies the mismatch between the actual angular response of a radiometer and the ideal behavior expected, given by the cosine law. (Line 127-129)

Reviewer 2 Report

This is a well written and executed study, investigating the accuracy of UV measurements with devices of a range of prices over a year in a tropical region. This work is potentially useful for a wide variety of industries. While generally well presented and thorough I have the following comments:

  1. English can be improved in places, there are several examples of words missing/incorrect phrasing and tensing.
  2. Many abbreviations are not defined within the text
  3. In the introduction you state UVI can be ‘UVI can be twice the extreme UVI value of 11’ are you suggesting it is at 22? Later in the methods, you state you had readings up to 16, so it might be worth adding the average UVI in addition to the highest reported (and make clearer this is 22).
  4. You state the Bentham was calibrated every 3 months – what about the test devices? Is this a fair comparison in the effectiveness if the gold standard method (the Bentham) is calibrated and the test devices are not? This should be discussed. If this is not the case and I am mistaken, please reflect in the text.
  5. In the results and discussion section, there are a lot of results and not much discussion. This needs to be expanded significantly as there is much to discuss and comment on. For example, is there any comment that can be made about the accuracy of using these devices at different latitudes (based on Zenith angle calculations) and places with lower UV exposure based on the data at different times of the year/day? There is more discussion in addition to this that is possible.
  6. It would be good to perform some calculations to show how different the UVI could be with each device under different conditions and over different time periods e.g. over a day/week/month/year of measurements the total UVI could differ with each device by X compared to the Bentham device. This would give readers a simpler understanding of the limitations of the devices. While your analysis is thorough, simply knowing if a device may be ‘good enough’ based on UVI readings with a certain degree of accuracy for someone’s needs would be useful.
  7. The conclusions section is very brief and should be expanded in line with the discussion.

Author Response

Dear Reviewer,

First of all, I would like to thank you for your work on our paper. Our response is structured based on your review. We identified 7 comments, referred hereafter from C1 to C7. Our responses are labelled accordingly from R1 to R7. To make the reading of the revised manuscript easier, all edits are highlighted in yellow.

With kind regards,

Jean-Maurice CADET

This is a well written and executed study, investigating the accuracy of UV measurements with devices of a range of prices over a year in a tropical region. This work is potentially useful for a wide variety of industries. While generally well presented and thorough I have the following comments:

C1. English can be improved in places, there are several examples of words missing/incorrect phrasing and tensing.

R1. We thank the Reviewer for the comment. The manuscript has been read carefully to improve the English throughout.

C2. Many abbreviations are not defined within the text

R2. All abbreviations are defined in the revised version of the manuscript.

C3. In the introduction you state UVI can be ‘UVI can be twice the extreme UVI value of 11’ are you suggesting it is at 22? Later in the methods, you state you had readings up to 16, so it might be worth adding the average UVI in addition to the highest reported (and make clearer this is 22).

R3. For more clarification, the sentence in lines 61-62 was rephrased.

C4. You state the Bentham was calibrated every 3 months – what about the test devices? Is this a fair comparison in the effectiveness if the gold standard method (the Bentham) is calibrated and the test devices are not? This should be discussed. If this is not the case and I am mistaken, please reflect in the text.

R4. KZ and SL were calibrated by PMOD/WRC before the inter-comparison campaign. SG and DV were calibrated by the manufacturer (mentioned in the manuscript)

However, the purpose of the study is not to perform a fair test between BT and KZ, SL, SG and DV. In fact, we do not know under what conditions these low- and moderate-cost devises will be used, or whether the owners will be able to calibrate them or not. The important fact about this study is to use the BT, which is regularly calibrated, as the reference instrument, in order to evaluate their performances.

C5. In the results and discussion section, there are a lot of results and not much discussion. This needs to be expanded significantly as there is much to discuss and comment on. For example, is there any comment that can be made about the accuracy of using these devices at different latitudes (based on Zenith angle calculations) and places with lower UV exposure based on the data at different times of the year/day? There is more discussion in addition to this that is possible.

R5. A paragraph has been added in the subsection (3.3) about the use of these sensors in different latitude sites (Lines 355-359).

C6. It would be good to perform some calculations to show how different the UVI could be with each device under different conditions and over different time periods e.g. over a day/week/month/year of measurements the total UVI could differ with each device by X compared to the Bentham device. This would give readers a simpler understanding of the limitations of the devices. While your analysis is thorough, simply knowing if a device may be ‘good enough’ based on UVI readings with a certain degree of accuracy for someone’s needs would be useful.

R6. Indeed, it would have been important to conduct the inter-comparison campaign in a different location and for different periods of time. We have included this as a study limitation and suggestion for further research in the manuscript. We did use the RD parameter, in addition to other statistics such as RSD and RSME, and that could help to answer part of C6. Indeed, we believe that RD, RSD and RSME values (see Table-1 and Figure 8) could give a simple and global understanding of the performance of each device.

C7. The conclusions section is very brief and should be expanded in line with the discussion.

R7. Conclusion paragraph was expanded.

Reviewer 3 Report

The authors present an interesting and well-designed study comparing moderate and low-cost spectrometers with a high-quality device used as a reference. Authors chose an excellent study location representing very high UV exposure levels in an isolated region that limits confounding variables and tests the instruments methodically. The paper is overall well written and referenced. This information could be useful to areas that wish to perform reliable ground-based UV monitoring, particularly in areas having high UV levels, with budgetary restrictions. Figures are informative, constructed well, and labelled thoroughly. Although my comments a generally favorable, I do have some specific comments as outlined below.

  1. Line 42: Temporal drift is an item of considerable concern and is mentioned in the abstract here and in the conclusions of the present paper (lines 388-389). However, the abstract mentions ‘temporal drift’ and ‘stability over time.’ Use of both terms is confusing as temporal drift and stability over time refer to the same thing. I recommend eliminating the phrase ‘stability over time’ in the abstract; consider revision of the statement in lines 388-389.
  2. Lines 91-93: Each of the models tested includes a cost estimate with the exception of the reference model. Please add approximate cost for BT if possible, and comment on availability or lack thereof. Also, readers may be interested in potential additional considerations such as cost of maintenance and frequency of calibration required for the instruments.

Author Response

Dear Reviewer,

First of all, I would like to thank you for your work on our paper. Our response is structured based on your review. We identified two comments, referred hereafter as C1 and C2. Our responses are labelled R1 and R2. To make the reading of the revised manuscript easier, all edits are highlighted in yellow.

With kind regards,

Jean-Maurice CADET

The authors present an interesting and well-designed study comparing moderate and low-cost spectrometers with a high-quality device used as a reference. Authors chose an excellent study location representing very high UV exposure levels in an isolated region that limits confounding variables and tests the instruments methodically. The paper is overall well written and referenced. This information could be useful to areas that wish to perform reliable ground-based UV monitoring, particularly in areas having high UV levels, with budgetary restrictions. Figures are informative, constructed well, and labelled thoroughly. Although my comments a generally favorable, I do have some specific comments as outlined below.

C1. Line 42: Temporal drift is an item of considerable concern and is mentioned in the abstract here and in the conclusions of the present paper (lines 388-389). However, the abstract mentions ‘temporal drift’ and ‘stability over time.’ Use of both terms is confusing as temporal drift and stability over time refer to the same thing. I recommend eliminating the phrase ‘stability over time’ in the abstract; consider revision of the statement in lines 388-389.

R1. Thank you for your comment. We have eliminated the phrase ‘stability over time’ in the abstract and the sentence lines 404-406 has been rephrased.

C2. Lines 91-93: Each of the models tested includes a cost estimate with the exception of the reference model. Please add approximate cost for BT if possible, and comment on availability or lack thereof. Also, readers may be interested in potential additional considerations such as cost of maintenance and frequency of calibration required for the instruments.

R2. Information about the Bentham cost has now been embedded in the revised version (see lines 112-114).
